# The impact of varying the number and selection of conditions on estimated multimorbidity prevalence: A cross-sectional study using a large, primary care population dataset

Clare MacRae[1,2]*, Megan McMinn[2], Stewart W. Mercer[1,2], David Henderson[2], David A. McAllister[3], Iris Ho[1,2], Emily Jefferson[4], Daniel R. Morales[4], Jane Lyons[5], Ronan A. Lyons[5], Chris Dibben[6], Bruce Guthrie[1,2]

1 The Advanced Care Research Centre, Usher Institute of Population Health Sciences and Informatics, University of Edinburgh, Edinburgh, United Kingdom, 2 The Usher Institute of Population Health Sciences and Informatics, University of Edinburgh, Edinburgh, United Kingdom, 3 University of Glasgow Institute of Health and Wellbeing, Glasgow, United Kingdom, 4 University of Dundee Division of Population Health and Genomics, Dundee, United Kingdom, 5 Swansea University Medical School, Swansea, United Kingdom, 6 The University of Edinburgh School of GeoSciences, Edinburgh, United Kingdom

* clare.macrae@ed.ac.uk

**Data Availability Statement:** All relevant data are within the manuscript or available via GitHub repository at https://github.com/macraec. Raw data cannot be made publicly available to protect data security. The third-party contact for access to

## Abstract

### Background

Multimorbidity prevalence rates vary considerably depending on the conditions considered in the morbidity count, but there is no standardised approach to the number or selection of conditions to include.

### Methods and findings

We conducted a cross-sectional study using English primary care data for 1,168,260 participants who were all people alive and permanently registered with 149 included general practices. Outcome measures of the study were prevalence estimates of multimorbidity (defined as $\geq 2$ conditions) when varying the number and selection of conditions considered for 80 conditions. Included conditions featured in $\geq 1$ of the 9 published lists of conditions examined in the study and/or phenotyping algorithms in the Health Data Research UK (HDR-UK) Phenotype Library. First, multimorbidity prevalence was calculated when considering the individually most common 2 conditions, 3 conditions, etc., up to 80 conditions. Second, prevalence was calculated using 9 condition-lists from published studies. Analyses were stratified by dependent variables age, socioeconomic position, and sex. Prevalence when only the 2 commonest conditions were considered was 4.6% (95% CI [4.6, 4.6] $p < 0.001$), rising to 29.5% (95% CI [29.5, 29.6] $p < 0.001$) considering the 10 commonest, 35.2% (95% CI [35.1, 35.3] $p < 0.001$) considering the 20 commonest, and 40.5% (95% CI [40.4, 40.6] $p < 0.001$) when considering all 80 conditions. The threshold number of conditions at which multimorbidity prevalence was >99% of that measured when considering all 80 conditions

raw data is CPRD's Research Data Governance (RDG) Process at rdg@cprd.com.

**Funding:** This work was supported by the Chief Scientist Office (HIPS/18/30) to BG, SWM, DA, EJ, DM, NHS Education for Scotland Academic Fellowship for CM, and Medical Research Council MR/W000253/1 fellowship for CM. This study/project is funded by the National Institute for Health Research (NIHR) Artificial Intelligence and Multimorbidity: Clustering in Individuals, Space and Clinical Context (AIM-CISC) grant NIHR202639. The funders had no role in study design, data collection and analysis, decision to publish, or preparation of the manuscript.

**Competing interests:** The authors have declared that no competing interests exist.

**Abbreviations:** CPRD, Clinical Practice Research Datalink; GI, gastro-intestinal; HDR-UK, Health Data Research UK; IMD, Index of Multiple Deprivation; IQR, interquartile range; RR, relative risk; SEP, socioeconomic position.

was 52 for the whole population but was lower in older people (29 in >80 years) and higher in younger people (71 in 0- to 9-year-olds). Nine published condition-lists were examined; these were either recommended for measuring multimorbidity, used in previous highly cited studies of multimorbidity prevalence, or widely applied measures of "comorbidity." Multimorbidity prevalence using these lists varied from 11.1% to 36.4%. A limitation of the study is that conditions were not always replicated using the same ascertainment rules as previous studies to improve comparability across condition-lists, but this highlights further variability in prevalence estimates across studies.

## Conclusions

In this study, we observed that varying the number and selection of conditions results in very large differences in multimorbidity prevalence, and different numbers of conditions are needed to reach ceiling rates of multimorbidity prevalence in certain groups of people. These findings imply that there is a need for a standardised approach to defining multimorbidity, and to facilitate this, researchers can use existing condition-lists associated with highest multimorbidity prevalence.

---

## Author summary

### Why was this study done?

- There is wide variety in the conditions considered by researchers when measuring multimorbidity prevalence.

- A systematic review of 566 studies, published in 2021, found lack of consensus in the selection of conditions considered.

- In half of studies only 8 conditions (diabetes, stroke, cancer, chronic obstructive pulmonary disease, hypertension, coronary heart disease, chronic kidney disease, and heart failure) were consistently considered, and the number of conditions considered varied from 2 to 285 (median 17).

- A more consistent approach to measuring multimorbidity is needed to facilitate comparability and generalisability across studies.

### What did the researchers do and find?

- We examined the impact of varying the conditions considered when measuring multimorbidity prevalence. We combined different numbers of conditions (from a list of 80) and selections of conditions (using 9 published condition-lists used to define and measure comorbidity, multimorbidity, and its prevalence) to determine how multimorbidity prevalence changed. All conditions were counted in the same way using publicly available code lists.

- There are large differences in prevalence, a range of 4.6% to 40.5%, when different numbers and selections of conditions are considered.

- People who are the oldest, living in the most deprived areas, and men require fewer conditions to be considered to reach close to multimorbidity prevalence when considering all 80 conditions (the ceiling effect, where the prevalence approaches the upper limit of prevalence possible in the study).

- Highest multimorbidity prevalence was found when using the Ho always + usually (derived from a recent Delphi consensus study), Barnett (widely used to measure multimorbidity prevalence), and Fortin (recommended for use in measuring multimorbidity) condition-lists.

### What do these findings mean?

- There is a need for standardisation when measuring multimorbidity prevalence so that results across studies are comparable and population subgroups are accurately represented.

- To address this, researchers can consider using the Ho always + usually, Barnett, or Fortin condition-lists that report the highest and most stable estimates of multimorbidity prevalence (where adding further conditions to the count had very little impact).

## Introduction

Multimorbidity is increasing in prevalence due to improved survival from chronic diseases and population ageing, and now poses major challenges to healthcare systems worldwide [1]. Multimorbidity is common, increases substantially with advancing age, and is more common in women and people with lower socioeconomic position (SEP) [2,3]. Despite its importance, existing research literature is highly heterogenous in how it defines and measures multimorbidity [4]. Choice of conditions considered in the count (the denominator) when measuring multimorbidity prevalence is likely to be driven by pragmatic decision-making in the context of data availability [5], or by recycling of existing published condition-lists, and results in wide diversity in the number and selection of conditions considered in current multimorbidity literature [6]. In a systematic review of 566 studies of multimorbidity, the number of conditions considered in counts of multimorbidity prevalence ranged from 2 to 285 (median 17, interquartile range [IQR] 11 to 23) [4]. Only 8 core conditions were consistently considered in more than half of studies (diabetes, stroke, cancer, chronic obstructive pulmonary disease, hypertension, coronary heart disease, chronic kidney disease, and heart failure) [4].

As a result of this diversity, multimorbidity prevalence estimates vary widely across studies [7], making it difficult to make comparisons within the existing literature. Unsurprisingly, higher multimorbidity prevalence is reported by studies that consider a larger number of conditions in their count of multimorbidity [7,8], studies that consider conditions that are most prevalent [2], and in studies that include more people in older age groups [7]. The number and selection of conditions considered when measuring multimorbidity prevalence is therefore important, but there is little consistent guidance to support researchers when deciding which conditions to consider. Researchers have recommended condition-lists to consider in multimorbidity measurement, including 11 conditions by Diederichs and colleagues [6] and

20 conditions by Fortin and colleagues [9]. More recently, a modified Delphi panel study by Ho and colleagues [10] developed 2 condition-lists based on international consensus on the measurement of multimorbidity: one list recommending conditions to always consider and a second recommending conditions to usually consider when counting multimorbidity prevalence [10].

All multimorbidity research findings are dependent on decisions made at the earliest stages in measurement, including what has been measured, and therefore, building understanding of the properties of multimorbidity as a concept is needed. To inform researchers' choices, and improve the comparability and reproducibility of future research, it is important to understand the relationship between multimorbidity prevalence and the number and selection of conditions considered in the count. The aim of this study was to examine these relationships in a large primary care cohort.

## Methods

### Study design

A cross-sectional study design was used to examine the hypothesis that multimorbidity prevalence varies when considering different numbers of conditions, and different selections of conditions (using published recommended or commonly used condition-lists), in the count. The analyses were designed in November 2021, performed in February 2022, and no data-driven changes to analyses took place during this period. As part of the peer-review process, 1 additional condition-list was added [11], and the paper updated to include $p$-values as well as confidence intervals for proportions, and sensitivity analyses of variation in prevalence by deprivation and sex using raw rather than direct age-standardised data were added.

### Data sources

The study analysed routinely collected, anonymised individual-level data from English participants in the Clinical Practice Research Datalink (CPRD) Gold dataset, which are broadly representative of the United Kingdom population [12]. Available data included individual demography (age, SEP, and sex), clinical codes from both GP electronic health records (Read codes) and hospital admission data (ICD-10 codes), and laboratory results. SEP was defined as deciles of the English Index of Multiple Deprivation (IMD), a measure of relative deprivation according to small local area level, with deciles defined by national thresholds [13].

### Study participants

Study participants were all people who were alive and permanently registered with 149 included general practices on the study index date, November 30, 2015, with least 2 year's GP registration prior to this [14].

### Definition of variables

For each individual, we defined the presence of 80 conditions using a set of existing code lists that combined Read codes (version 2) applied to GP electronic health record data, International Classification of Diseases 10th version (ICD-10) codes applied to hospital admission data, and laboratory results recorded in the GP electronic health record [15] (S1 Table). The 80 conditions were chosen because they featured in 1 or more of the 9 published lists of conditions examined in the study and/or phenotyping algorithms (condition code lists) in the Health Data Research UK (HDR-UK) Phenotype Library [15]. New code lists were made by study authors for conditions featured in existing condition-lists where no HDR-UK algorithms

were available and are listed in S1 Table. All the codes used to identify individuals with each condition were mutually exclusive; therefore, double counting of conditions was not possible, and all conditions contained within condition-lists were included in the total 80 conditions.

Condition-lists were either specifically recommended for measuring multimorbidity (referred to hereinafter as Diederichs [6], Fortin [9], Ho always [10], Ho always + usually [10], and N'Goran [11]), used in previous highly cited large-scale studies of multimorbidity prevalence (Barnett [16] and Salive [17]), or included in widely applied measures of "comorbidity" (Charlson [18] and Elixhauser [19]). The 2 condition-lists recommended by the recent Ho and colleagues Delphi consensus study [10] were conditions recommended to always include (Ho always), and all the conditions recommended by both the lists, conditions to always include and to usually include (Ho always + usually) (S1 Information Panel).

Heterogeneity existed in the description and the hierarchical level of conditions between condition-lists. Therefore, to ascertain each condition in the same way for every condition-list, some were dis-aggregated to more granular descriptions. For example, Diederichs and colleagues [6] considered cancer, while Ho always [10] considered 3 condition groups that were all cancers: primary malignancy, secondary malignancy, and haematological malignancy. In this case, in the Diederichs and colleagues [6] condition-list, cancer was disaggregated from 1 to 3 conditions (to primary malignancy, secondary malignancy, and haematological malignancy) to allow direct comparison with Ho always [10]. Therefore, the number of included conditions in some condition-lists varied from the original published lists (S2 Table).

## Statistical analysis

Multimorbidity prevalence was calculated when different numbers and selections of conditions were considered in the count (the denominator). In all analyses, multimorbidity was defined by the cut-off (the numerator) that was the presence of $\geq 2$ conditions [3].

We conducted a suite of comparisons including examination of the effect of the number of conditions considered in the count on multimorbidity prevalence by considering the most common 2 conditions, followed by the most common 3 conditions, the most common 4 conditions, etc., for every number up to considering all 80 conditions in the count (S1 Information Panel). To do this, conditions were ordered from most to least prevalent and added in turn to each new count. The prop.test procedure in R was used to estimate 95% confidence intervals and ps for prevalence were calculated using the normal approximation for large samples.

By making the assumption that multimorbidity prevalence estimated by considering all 80 conditions in the count was the true prevalence, we then calculated the number of conditions that had to be included to exceed a relative risk (RR) of 0.99 of this "true" prevalence. This was done to estimate when a ceiling effect was present, where the prevalence approaches the upper limit of possible prevalence in the study and the point at which adding further conditions to the count had very little impact on multimorbidity prevalence.

To examine the effect the selection of conditions considered in the count, multimorbidity prevalence was calculated when considering the conditions included in each of the 9 condition-lists. Since age is very strongly associated with multimorbidity and the SEP and sex composition within different age groups varies making age a major confounder, analyses were standardised to the age structure of the whole study cohort as the standard population and age-specific standardised rates for population subgroups were calculated [20]. Sensitivity analysis was done using unstandardised rates.

This study adhered to the REporting of studies Conducted using Observational Routinely collected Data reporting guidelines [21] (S3 Table). All data management, statistical analyses, and plotting was done in R version 3.6.2 [22], available within in the ISO27001 and Scottish

Government approved Health Informatics Centre Safe Haven. The analysis was approved by CPRD Independent Scientific Advisory Committee (reference 20_018) under the terms of CPRD NHS Research Ethics dataset approval.

## Results

The study included 1,168,620 people. When considering all 80 conditions in the count, multimorbidity was present in 473,533 (40.5%) of the cohort. People with multimorbidity were older than the whole population, median 60 years (IQR 46 to 72) versus 44 years (IQR 23 to 60), more often women, 257,237 (54.3%) versus 587,687 (50.3%), and more often lived in the 5 most deprived IMD decile areas, 208,386 (44.0%) versus 505,322 (43.2%). Differences between the whole population and people with multimorbidity examined using $\chi^2$ tests for proportions within each age group, IMD decile, and both sexes, were statistically significant ($p < 0.001$) (Table 1).

**Table 1. Population characteristics.**

| | No. (% of total population/ each column) $N = 1,168,620$ | No. (% of each row) with multimorbidity when considering all 80 conditions in the count |
|---|---|---|
| Age group, years* | | |
| 0–9 | 113,955 (9.8) | 2,739 (2.4) |
| 10–19 | 137,517 (11.8) | 9,129 (6.6) |
| 20–29 | 122,237 (10.5) | 24,916 (20.4) |
| 30–39 | 143,243 (12.3) | 39,888 (27.8) |
| 40–49 | 176,061 (15.1) | 66,878 (37.9) |
| 50–59 | 173,435 (14.8) | 89,586 (51.5) |
| 60–69 | 141,041 (12.1) | 98,512 (69.9) |
| 70–79 | 97,843 (8.4) | 82,752 (84.4) |
| ≥80 | 63,288 (5.4) | 59,092 (93.4) |
| Sex* | | |
| Men | 580,933 (49.7) | 215,555 (37.1) |
| Women | 587,687 (50.3) | 257,049 (43.7) |
| IMD decile*, ** | | |
| 1 (most affluent) | 167,558 (14.0) | 62,032 (37.0) |
| 2 | 129,704 (11.0) | 51,504 (39.7) |
| 3 | 128,234 (11.0) | 51,794 (40.4) |
| 4 | 109,986 (9.4) | 45,681 (41.5) |
| 5 | 127,816 (11.0) | 53,601 (41.9) |
| 6 | 104,158 (8.9) | 44,279 (42.5) |
| 7 | 108,782 (9.3) | 44,097 (40.5) |
| 8 | 103,501 (8.9) | 43,102 (41.6) |
| 9 | 100,577 (8.6) | 40,019 (39.8) |
| 10 (most deprived) | 88,304 (7.6) | 36,495 (41.3) |

*There were statistically significant differences in the proportion of people with and without multimorbidity ($P < 0.001$) within each variable (row) group using $\chi^2$ tests.

**IMD.

IMD, Index of Multiple Deprivation.

Of the 80 conditions examined, 6 conditions were present in more than 5% of the whole population: hypertension was the most prevalent (9.1%), followed by depression (8.7%), asthma (7.6%), upper gastro-intestinal (GI) tract acid conditions (7.6%), anxiety (6.7%), and osteoarthritis (5.7%). Nine conditions were present in less than 0.1% of the population (Table 2).

There was marked variability in multimorbidity prevalence depending on the number of conditions considered in the count. Using all 80 conditions, multimorbidity prevalence was 40.5% (95% CI [40.4, 40.6] $p < 0.001$). When considering only the 2 conditions most prevalent in the whole population in the count, multimorbidity was present in 4.6% (95% CI [4.6, 4.6] $p < 0.001$) (Fig 1, S3 Table). When adding more conditions to the count (the most prevalent remaining conditions first), there was a steep increase in estimated multimorbidity prevalence, rising to 29.5% (95% CI [29.5, 29.6%] $p < 0.001$) when considering 10 conditions in the count. Following this, a more gradual increase in estimated prevalence was seen as more conditions were added to the count: 35.2% (95% CI [35.1, 35.3] $p < 0.001$) considering 20 conditions, and 37.4% (95% CI [37.3, 37.5] $p < 0.001$) considering 30 conditions. There was only 0.7 percentage point absolute difference in prevalence between considering 50 conditions, 39.8% (95% CI [39.7, 39.9] $p < 0.001$), and all 80 conditions (40.5%). In the whole population, the predefined ceiling where adding additional conditions had little impact on prevalence was reached at 52 conditions (i.e., estimated prevalence for 52 conditions versus 80 conditions RR >0.99).

Multimorbidity prevalence varied widely between the 9 different condition-lists, varying from 11.1% (95% CI [11.0, 11.2] $p < 0.001$) using the Ho always [10] list, to 36.4% (95% CI [36.3, 36.5] $p < 0.001$) using the Ho always + usually [10] list (Fig 1, Table 3). Three condition-lists (Ho always + usually [10], Barnett [16], and Fortin [9]) had prevalence close to that estimated by including the same number of the most common conditions in the number of conditions analysis (represented by proximity of these points to the black line in Fig 1). These lists also had highest RR of the multimorbidity prevalence calculated when considering all 80 conditions in the count: Ho always + usually [10] RR 0.90 (95% CI [0.89, 0.90] $p < 0.001$), Barnett and colleagues [16] RR 0.84 (95% CI [0.83, 0.84] $p < 0.001$), and Fortin and colleagues [9] RR 0.82 (95% CI [0.82, 0.82] $p < 0.001$) (Table 2). The remaining 5 condition-lists, however, had prevalence considerably below that estimated by including the same number of most common conditions with prevalence RR 0.27 (95% CI [0.27, 0.27] $p < 0.001$) and RR 0.27 (95% CI [0.27, 0.27] $p < 0.001$) respectively using the Ho always [10] and Charlson [18] condition-lists.

The initial gradient of increase in multimorbidity prevalence seen as conditions were added to the count was steepest in the oldest age groups, followed by flattening of the curve as more conditions were considered (Fig 2). In 0–9- and 10–19-year-olds, there was a more gradual increase in prevalence, because rarer conditions contribute to a higher proportion of multimorbidity in children and young people. The influence of adding additional numbers of conditions to the count on estimated prevalence plateaued at a lower number of conditions considered in older people. In people aged 80 years and over, the predefined ceiling (prevalence compared to 80 conditions RR >0.99) was reached at 29 conditions, compared to 71 conditions in those aged 0 to 9 years (Fig 2). In IMD-stratified analysis, there was a clear social gradient of multimorbidity prevalence irrespective of the number of conditions included, with the more deprived having higher prevalence than the less deprived (Fig 3). The predefined ceiling was reached at a lower number of conditions in the most deprived IMD decile (49 conditions) compared to the least deprived (54 conditions) (Fig 3). Sensitivity analysis using raw (unstandardised rates) had less clear social gradient (reflecting that the most deprived are on average younger than the affluent), and no clear pattern in the predefined ceiling across IMD deciles (S1 Fig).

**Table 2. Prevalence of individual conditions.**

| Prevalence rank | Long-term condition | Population count (%) N = 1,168,620 | Prevalence rank | Long-term condition | Population count (%) N = 1,168,620 |
|---|---|---|---|---|---|
| 1 | Hypertension | 212,520 (18.2) | 41 | Paroxysmal tachycardias | 8,747 (0.7) |
| 2 | Depression | 201,991 (17.3) | 42 | Obstructive and reflux uropathy | 8,501 (0.7) |
| 3 | Asthma | 177,301 (15.2) | 43 | Polymyalgia rheumatica | 8,446 (0.7) |
| 4 | Upper GI tract acid conditions | 176,518 (15.1) | 44 | Intellectual disability | 8,004 (0.7) |
| 5 | Anxiety | 156,762 (13.4) | 45 | Secondary malignancy | 7,332 (0.6) |
| 6 | Osteoarthritis | 132,799 (11.4) | 46 | Schizophrenia | 7,097 (0.6) |
| 7 | Primary malignancy | 74,917 (6.4) | 47 | Haematological malignancy | 7,028 (0.6) |
| 8 | Type 2 diabetes mellitus | 61,671 (5.3) | 48 | Autism | 6,714 (0.6) |
| 9 | Chronic kidney disease | 59,812 (5.1) | 49 | Visual impairment and blindness | 5,942 (0.5) |
| 10 | Coronary heart disease | 58,585 (5.0) | 50 | Obsessive compulsive disorder | 5,553 (0.5) |
| 11 | Thyroid disease | 57,289 (4.9) | 51 | Bronchiectasis | 5,123 (0.4) |
| 12 | Erectile dysfunction | 53,198 (4.5) | 52 | Bipolar affective disorder | 4,572 (0.4) |
| 13 | Diverticular disease | 46,410 (4.0) | 53 | Type 1 diabetes mellitus | 4,439 (0.4) |
| 14 | Urinary incontinence | 45,681 (3.9) | 54 | Coeliac disease | 4,188 (0.4) |
| 15 | Psoriasis | 41,871 (3.6) | 55 | Eating disorders | 3,766 (0.3) |
| 16 | Benign prostatic hyperplasia | 33,656 (2.7) | 56 | Tubulointerstitial nephropathy | 3,511 (0.3) |
| 17 | Gout | 33,483 (2.9) | 57 | Cardiomyopathy | 3,399 (0.3) |
| 18 | Atrial fibrillation | 33,098 (2.8) | 58 | Parkinson's disease | 2,938 (0.2) |
| 19 | Alcohol misuse | 31,841 (2.7) | 59 | Multiple sclerosis | 2,932 (0.2) |
| 20 | Chronic obstructive pulmonary disease | 31,654 (2.7) | 60 | Hyperparathyroidism | 2,316 (0.2) |
| 21 | Osteoporosis | 30,284 (2.6) | 61 | Ankylosing spondylosis | 2,214 (0.2) |
| 22 | Stroke and transient ischaemic attack | 27,925 (2.4) | 62 | Cerebral palsy | 2,051 (0.2) |
| 23 | Urolithiasis | 25,551 (2.2) | 63 | Lupus erythematosus | 1,752 (0.1) |
| 24 | Peripheral neuropathy | 24,065 (2.1) | 64 | Pulmonary fibrosis | 1,712 (0.1) |
| 25 | Hearing loss | 22,535 (1.9) | 65 | Primary thrombocytopaenia | 1,707 (0.1) |
| 26 | Venous thromboembolic disease | 21,058 (1.8) | 66 | Giant cell arteritis | 1,662 (0.1) |
| 27 | Epilepsy | 19,139 (1.6) | 67 | Paralysis | 1,625 (0.1) |
| 28 | Heart valve disease | 18,172 (1.5) | 68 | Primary pulmonary hypertension | 1,210 (0.1) |
| 29 | Heart failure | 17,879 (1.5) | 69 | Sjogren syndrome | 1,184 (0.1) |
| 30 | Substance misuse | 17,433 (1.5) | 70 | Sick sinus syndrome | 1,097 (0.1) |
| 31 | Inflammatory arthritis | 15,853 (1.4) | 71 | Thalassaemia | 1,088 (0.1) |
| 32 | Endometriosis | 14,567 (1.2) | 72 | Human immunodeficiency virus | 960 (0.1) |
| 33 | Sleep apnoea | 12,853 (1.1) | 73 | Diabetes mellitus other or not specified | 933 (0.1) |
| 34 | Raynaud's disease | 12,312 (1.0) | 74 | Aplastic anaemia | 820 (0.1) |
| 35 | Neuropathic bladder | 11,507 (1.0) | 75 | Cystic fibrosis | 559 (0.05) |
| 36 | Dementia | 11,374 (1.0) | 76 | Scleroderma | 442 (0.04) |
| 37 | Inflammatory bowel disease | 10,949 (0.9) | 77 | Myasthenia gravis | 419 (0.04) |
| 38 | Liver disease | 10,628 (0.9) | 78 | Sickle cell disease | 415 (0.04) |
| 39 | Peripheral arterial disease | 10,427 (0.9) | 79 | Addison's disease | 394 (0.03) |
| 40 | Heart block and bundle branch block | 10,231 (0.9) | 80 | Motor neurone disease | 231 (0.02) |

GI, gastro-intestinal.

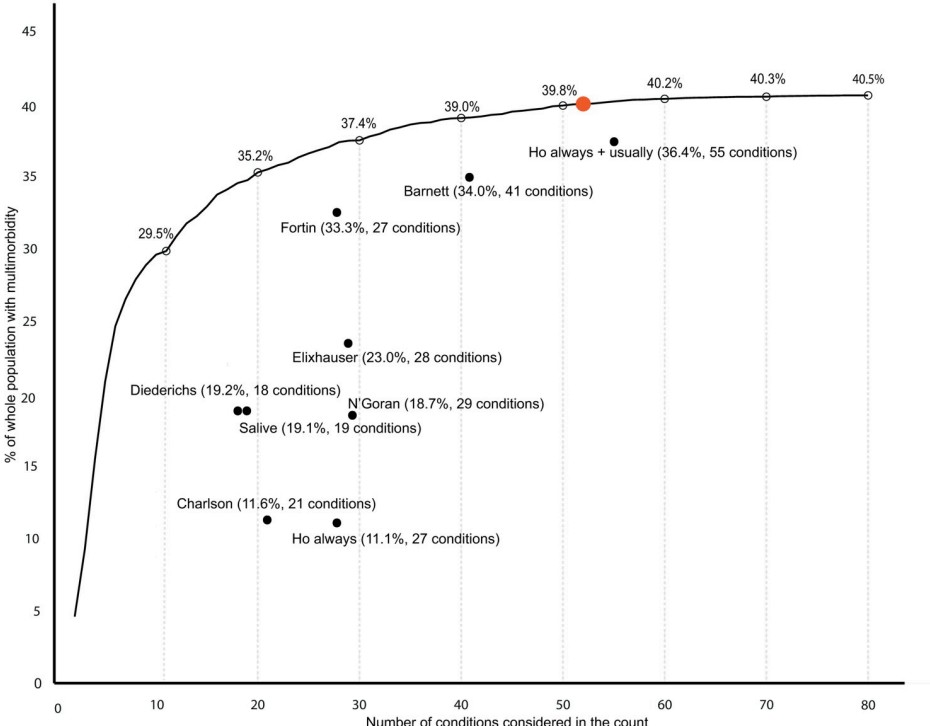

**Fig 1. Multimorbidity prevalence according to number of conditions, the ceiling effect where adding additional conditions had little impact on prevalence, and selection of conditions using existing condition-lists.** The black line represents multimorbidity prevalence calculated when considering different numbers of conditions in the count ranging from 2 to all 80 conditions, where conditions were added in order of most to least prevalent (e.g., at 2 conditions this is multimorbidity prevalence considering the most common 2 conditions). Percentage prevalence of multimorbidity when 10, 20, 30, 40, 50, 60, 70, and 80 conditions were considered is marked at empty black circles above the black line. The number of conditions at which RR was >0.99 of multimorbidity prevalence of having the same multimorbidity prevalence when all 80 conditions were considered (ceiling effect) was reached is marked with an orange dot (at 52 conditions). Black dots represent multimorbidity prevalence when considering conditions included in existing condition-lists and are annotated with the condition-list name, prevalence, and number of conditions considered. RR, relative risk.

**Table 3. Multimorbidity prevalence using existing condition-lists and RR of multimorbidity when considering all 80 conditions (ceiling effect where adding additional conditions had little impact on prevalence).**

| Condition-list | No. of conditions in condition-list | Multimorbidity prevalence No. (%) | RR (95% CI) |
|---|---|---|---|
| All 80 conditions (reference) | 80 | 473,533 (40.5) | 1.0 |
| Ho always + usually [10] | 55 | 425,413 (36.4) | 0.90 (0.89–0.90) |
| Barnett [16] | 41 | 397,009 (34.0) | 0.84 (0.83–0.84) |
| N'Goran [11] | 29 | 219,098 (18.7) | 0.46 (0.46–0.46) |
| Elixhauser [19] | 28 | 268,261 (23.0) | 0.57 (0.56–0.57) |
| Fortin [9] | 27 | 389,286 (33.3) | 0.82 (0.82–0.82) |
| Ho always [10] | 27 | 129,698 (11.1) | 0.27 (0.27–0.27) |
| Charlson [18] | 21 | 135,166 (11.6) | 0.28 (0.28–0.29) |
| Diederichs [6] | 18 | 224,001 (19.2) | 0.47 (0.47–0.47) |
| Salive [17] | 14 | 222,859 (19.1) | 0.47 (0.47–0.47) |

RR, relative risk.

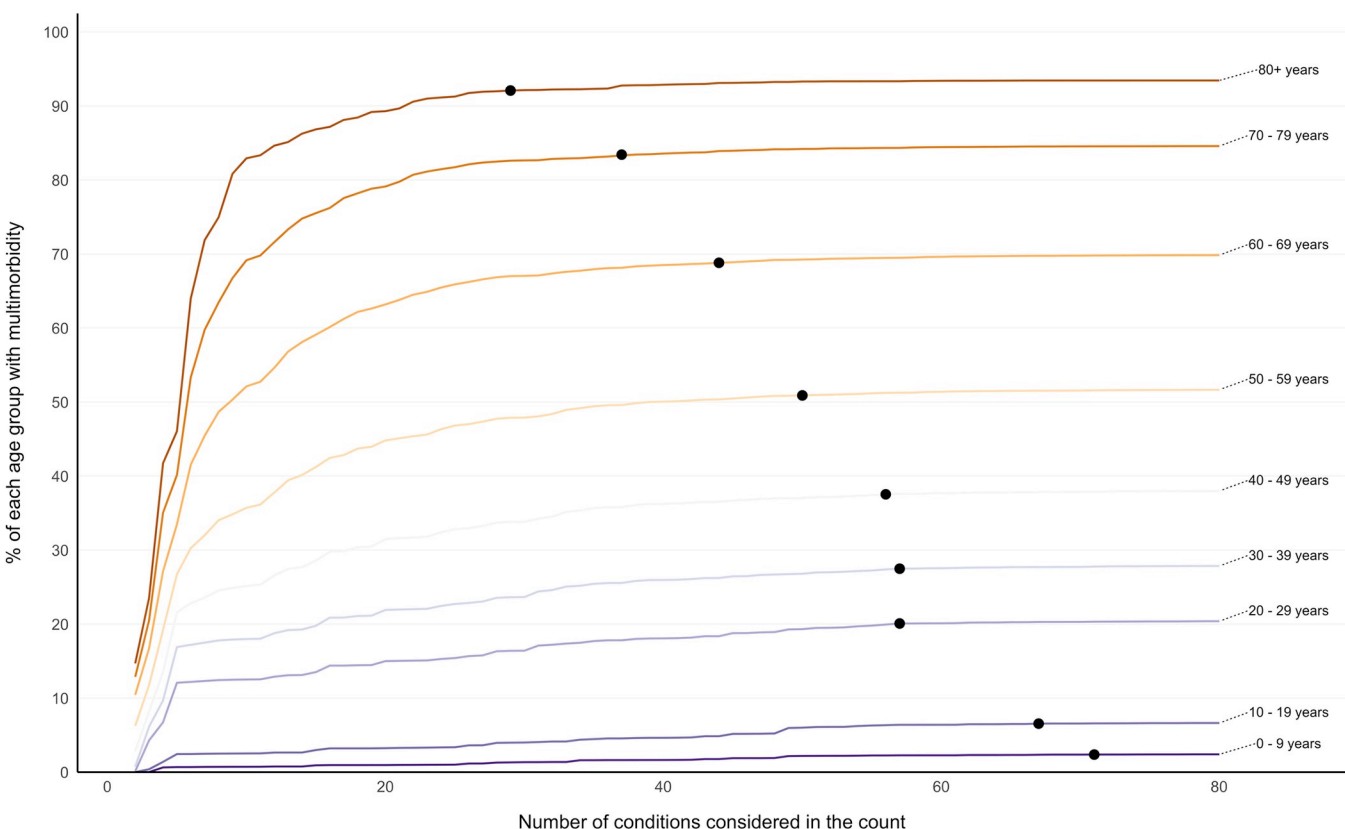

**Fig 2. Age-stratified multimorbidity prevalence according to number of conditions considered, reporting the ceiling effect where adding additional conditions had little impact on prevalence.** Labelled coloured lines represent multimorbidity prevalence calculated when considering different numbers of conditions in the count ranging from 2 to all 80 conditions stratified into age groups. Black dots represent the number of conditions at which RR >0.99 of multimorbidity prevalence of having the same multimorbidity prevalence when all 80 conditions were considered (ceiling effect): 0–9 years at 71 conditions, 10–19 years at 67 conditions, 20–29 conditions at 57 conditions, 30–39 years at 57 conditions, 40–49 years at 56 conditions, 50–59 years at 50 conditions, 60–69 years at 44 conditions, 70–79 years at 37 conditions, 80+ years at 29 conditions. RR, relative risk.

Multimorbidity prevalence was higher in women and girls at every level of number of conditions in the direct age-standardised analysis, and the predefined ceiling was reached at a higher number of conditions in women and girls than in men and boys (Fig 4). In the sensitivity analysis using unstandardised rates, there was a larger gap in multimorbidity prevalence between sexes, reflecting that women are on average older, and the predefined ceiling was reached at a similar number of conditions to the direct age-standardised analysis (S2 Fig).

In both age and deprivation stratified analyses, fewer conditions were required to reach RR 0.99 for the groups with highest prevalence. However, a different pattern was seen in men (who had lower multimorbidity prevalence) where the ceiling was reached at 50 conditions, compared to 54 in women (Fig 4).

The age distribution of multimorbidity prevalence was not uniform across the 9 condition-lists. Across all ages, multimorbidity prevalence using the Ho always + usually [10] condition-list was closest to prevalence when considering all 80 conditions (Fig 5). The Fortin [9], Barnett [16], and Elixhauser [19] condition-lists had lower prevalence than Ho always + usually [10] but followed a similar upward trajectory from youngest to oldest. Salive and colleagues [17] and Diederichs and colleagues [6] had low prevalence in younger age groups, but multimorbidity prevalence increased steeply from age 50 to 59 years and older onwards. Ho always [10] and Charlson [18] had markedly lower prevalence rates than other condition-lists across all age groups.

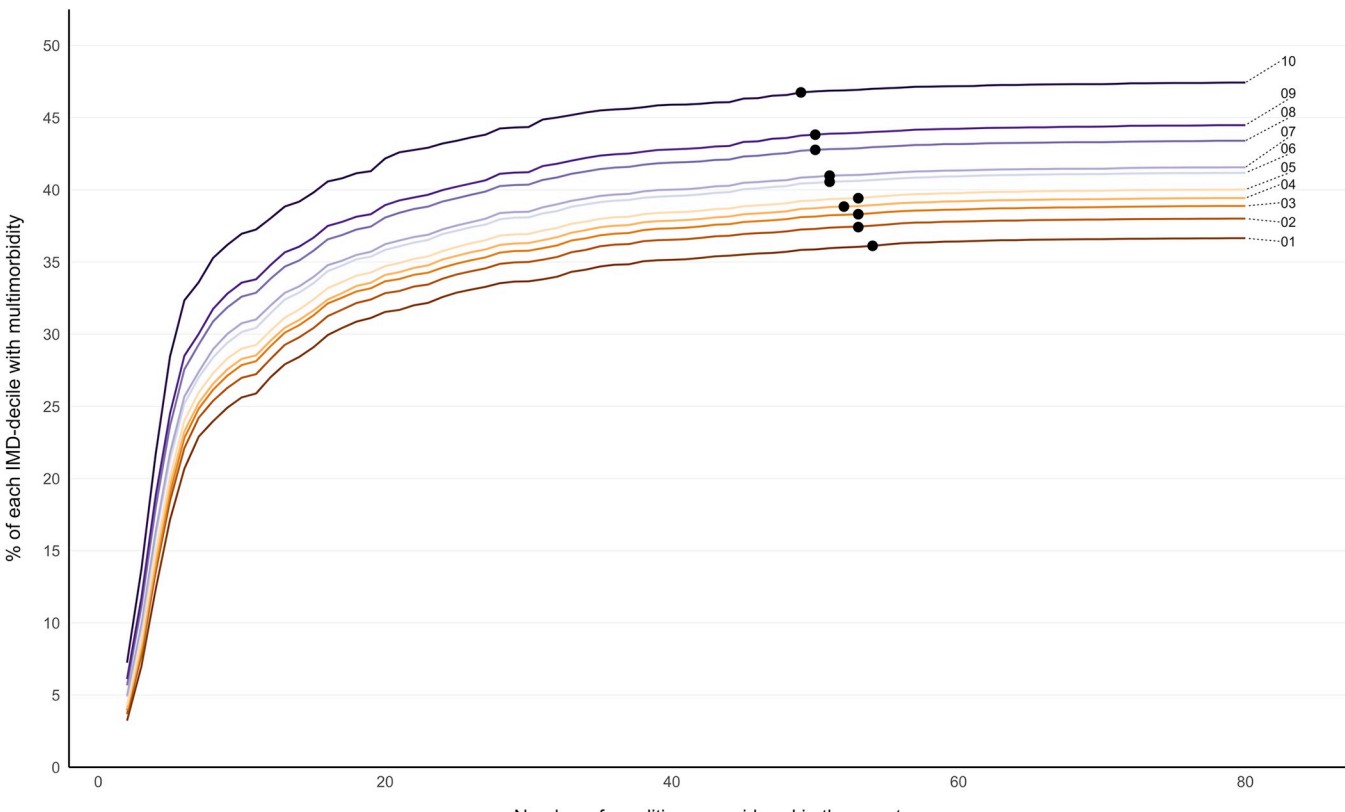

**Fig 3. SEP-stratified multimorbidity prevalence according to number of conditions considered following direct age standardisation, reporting the ceiling effect where adding additional conditions had little impact on prevalence.** Labelled coloured lines represent multimorbidity prevalence calculated when considering different numbers of conditions in the count ranging from 2 to all 80 conditions stratified into IMD deciles where IMD 1 is least and IMD 10 is most deprived. Black dots represent the number of conditions at which RR >0.99 of multimorbidity prevalence of having the same multimorbidity prevalence when all 80 conditions were considered (ceiling effect): IMD 10 at 49 conditions, IMD 9 at 50 conditions, IMD 8 at 50 conditions, IMD 7 at 51 conditions, IMD 6 at 51 conditions, IMD 5 at 53 conditions, IMD 4 at 52 conditions, IMD 3 at 53 conditions, IMD 2 at 53 conditions, and IMD 1 at 54 conditions. Direct age standardisation where the whole study cohort was the standard population was applied (see S1 Fig for unstandardised rates). IMD, Index of Multiple Deprivation; RR, relative risk; SEP, socioeconomic position.

## Discussion

This study found very large differences in estimated multimorbidity prevalence from varying the number and selection of conditions considered in the count, and in younger people, the more affluent, and women, including additional relatively rare conditions had larger impact on estimated multimorbidity prevalence. Multimorbidity prevalence differed considerably by varying the number of conditions, ranging from 4.6% to 40.5%, and selection of conditions considered, ranging from 11.1% to 36.4% using 9 previously published lists of conditions [6,9,10,16–19]. Counting multimorbidity prevalence using the 9 existing condition-lists resulted in lower estimated prevalence than when considering the same number of the most common conditions, although the extent of this varied: Ho always + usually [10], Fortin [9], and Barnett [16] had the best performance.

Consistent with the findings of this study, there is a wide range in the number and selection of conditions considered in the current multimorbidity literature [4], and in estimates of multimorbidity prevalence [7]. A systematic review of 566 multimorbidity studies by Ho and colleagues [4] found that the number of conditions considered by existing studies ranged from 2 to 285 (median 17, IQR 11 to 23), and very little uniformity in terms of the selection of

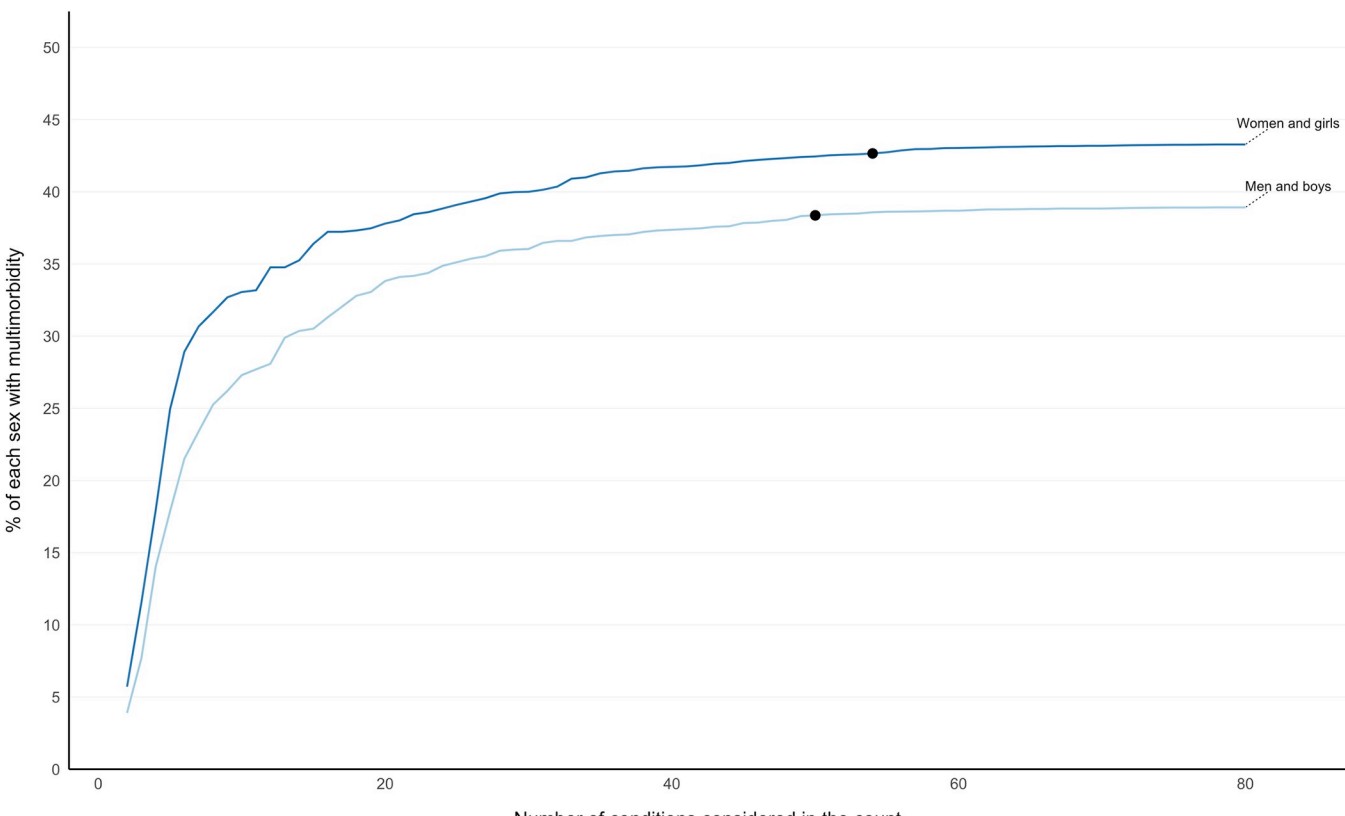

**Fig 4. Sex-stratified multimorbidity prevalence according to number of conditions considered following direct age standardisation, reporting the ceiling effect where adding additional conditions had little impact on prevalence.** Labelled coloured lines represent multimorbidity prevalence calculated when considering different numbers of conditions in the count ranging from 2 to all 80 conditions stratified by sex. Black dots represent the number of conditions at which RR >0.99 of multimorbidity prevalence of having the same multimorbidity prevalence when all 80 conditions were considered (ceiling effect): women and girls at 54 conditions and men and boys at 50 conditions. Direct age standardisation where the whole study cohort was the standard population was applied (see S2 Fig for unstandardised rates). RR, relative risk.

conditions was found across studies. Only 8 conditions (diabetes, stroke, cancer, chronic obstructive pulmonary disease, hypertension, coronary heart disease, chronic kidney disease, and heart failure) were considered in at least half of the studies, and a quarter of studies did not consider any mental health condition. Simard and colleagues [23] reviewed existing literature to examine how studies used, developed, and validated methods for measuring multimorbidity. They found heterogeneity in the grouping of conditions, validation processes, number of ICD-10 code digits used to define included conditions, and use of additional data sources. Diederichs and colleagues [6] recognised the need to establish a standardised instrument to measure multimorbidity and recommended a minimum set of 11 conditions to include (cancer, diabetes mellitus, depression, hypertension, myocardial infarction, chronic ischemic heart disease, heart arrhythmias, heart insufficiency, stroke, COPD, and arthritis). These conditions were selected based on high prevalence and a severe impact on affected individuals in terms of impairment of function and high need for management, from a population of people aged over 64 years old in Germany. A recent systematic review and meta-analysis of 193 studies examining multimorbidity prevalence [7] did not directly compare prevalence when considering different condition-lists, however, did find that prevalence was significantly higher in studies considering a larger number of conditions in the count: studies considering 44 or more

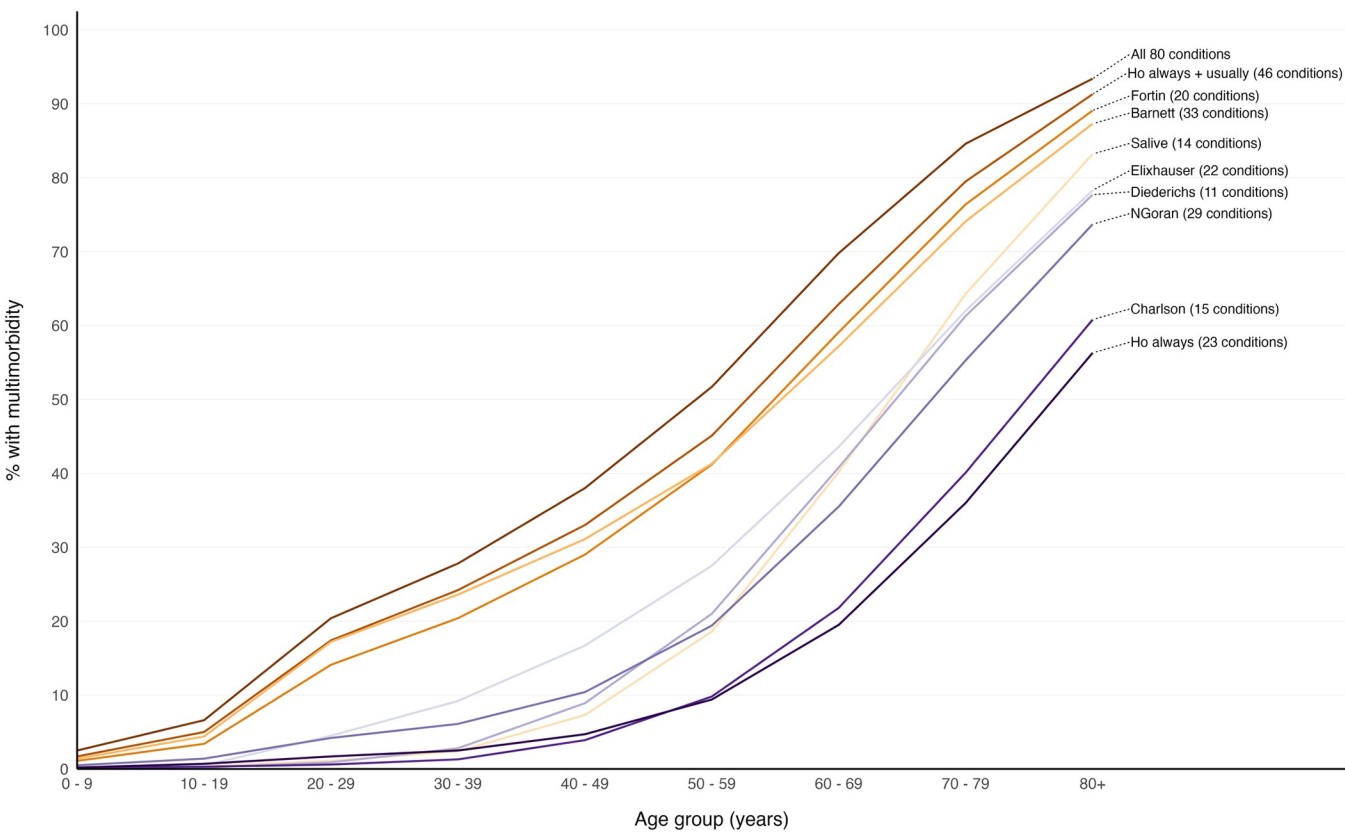

**Fig 5. Multimorbidity prevalence by age considering all 80 conditions and according to existing condition-lists.** Labelled coloured lines represent multimorbidity prevalence calculated for each age group when considering conditions in each condition-list.

conditions had higher pooled multimorbidity prevalence (87.6%) than studies considering fewer than 9 conditions (30.1%).

Strengths of this study include comprehensive analysis of multimorbidity prevalence estimates in a large population dataset derived from primary care electronic health records. Analysis systematically examined multimorbidity prevalence in the same population for different numbers of conditions considered in the count and using condition-lists recommended or used in previous studies. However, a limitation is that we did not necessarily replicate how previous studies ascertained the presence of conditions, but instead defined the presence of each condition using published UK code lists. This improves comparability within this study but highlights that further variability in prevalence estimates will happen because of variation in how each condition is ascertained (i.e., variation in exactly which codes or prescriptions are used, or restrictions on how recent a diagnosis must be). There is heterogeneity in which conditions are included between existing lists of conditions, and therefore, decisions were made about how to standardise conditions to the full list of 80 conditions. For example, the Barnett (2012) [16] condition-list used time-limited diagnoses that were not replicated in this study in order to make condition ascertainment consistent across the condition-lists examined. Using electronic health records to ascertain the prevalence of conditions can be associated with under ascertainment because the absence of a record does not necessarily mean absence of the condition, and more severe disease is likely to be overrepresented in medical records [24]. The data were from 2015, however, prevalence of the commonest condition hypertension was unchanged between 1990 and 2019 in a pooled analysis of worldwide population studies (32%

of women in 2019 versus 32% in 1990) [25], and this study identified similar prevalence rates of depression as ascertained by an Office of National Statistics survey from 2021 [26].

Deciding which conditions to include in multimorbidity research is complicated, including in the extent to which conditions should be aggregated (e.g., coronary heart disease) or considered separately (e.g., angina, myocardial infarction). Ideally, researchers would use a standardised list to improve research comparability and reproducibility, but this is not always feasible due to varying data availability and varying prevalence of disease in different settings. An alternative method is to use an "open condition-list," as used by Fortin and colleagues [8] in a Canadian study where methodology was not constrained to a specified number of conditions considered in the count to calculate multimorbidity prevalence, but considered all conditions present in a patient's medical records in the multimorbidity count. The number of conditions in study participants was highly variable and resulted in large differences in multimorbidity prevalence, particularly for younger people [7]. The method of data collection involved manual review of medical records, and therefore, although this analysis provides additional richness, it would be challenging to scale this approach and apply it to larger populations.

Even where researchers agree on which conditions to measure, there is an additional source of variation introduced by heterogeneity in methods chosen to measure and ascertain those conditions in data. Based on this research, if the purpose of the study is to estimate prevalence then estimates will be relatively stable providing the 50 most common conditions are considered; however, this threshold requires examination in other datasets and settings. Although some tailoring to local context and purpose will often be necessary, comparability and reproducibility would be improved by choices always starting with a core list of conditions. Researchers should therefore consider using the Delphi consensus derived Ho always + usually list [10], or for measuring prevalence the Barnett [16], or Fortin [9] condition-lists.

There are several areas where further research is needed. First, this study examined relationships between the number and selection of conditions and multimorbidity prevalence in the UK. However, similar studies in low- and middle-income countries are needed, where prevalence of individual conditions will be different. Second, condition ascertainment in routine data is based on lists of clinical codes (and sometimes prescribing or laboratory data) [27]. However, there can be large variations in the clinical codes used to define the same condition in different studies [28]. Therefore, further exploration of the impact of variation in which codes or prescriptions are used to define conditions is needed. Applying the condition codes from validated open-access published code lists, such as the HDR-UK Phenotype Library [15] or other similar sources [29] will also improve comparability and reproducibility. Third, it is important to examine whether and how much the number and selection of conditions considered in counts alter observed associations with important clinical outcomes such as functional status, unplanned hospital use, and death.

The key implication of this study is that the choice of conditions to consider when estimating multimorbidity prevalence has a large impact on the results, with additional variation in impact between older versus younger people particularly. The comparability and reproducibility of multimorbidity research would be improved by researchers including recommended core conditions wherever possible [10], with explicitly justified variation for study context and purpose.

## Supporting information

**S1 Table. List of chronic conditions examined and codes used to define those morbidities.** (DOCX)

**S2 Table. Conditions in each condition-list as implemented in this study, and as stated in published condition-list.**
(DOCX)

**S3 Table. The RECORD statement.** Checklist of items, extended from the STROBE statement, that should be reported in observational studies using routinely collected health data.
(DOCX)

**S1 Information Panel. Measuring multimorbidity in research: a Delphi consensus study.** Two recommended condition-lists defined by a modified Delphi panel study [10] were used. This study developed international consensus on the measurement of multimorbidity in research and was funded by HDR-UK. Data were collected in 3 rounds of online questionnaires, including 25 public panel and 150 professional panel members. Public members had an interest in, or personal experience of, multimorbidity. Professional participants were clinicians, policy makers, and researchers interested in, or involved in, multimorbidity work. Two sets of questions were developed separately for the public and professional panels. Questions in subsequent rounds were informed by results from the previous questionnaire. Participants were asked to answer open and closed questions, where open questions were subsequently triangulated by subsequent closed questions. Consensus was reached for 24 conditions to always include in multimorbidity measures, and 35 conditions to usually include unless a good reason not to, and these lists have been examined in our study of multimorbidity prevalence. In the study, we calculated multimorbidity prevalence using the always include list (Ho always) and both condition-lists together (Ho always + usually).
(EPS)

**S1 Fig. Socioeconomic position stratified multimorbidity prevalence according to number of conditions considered without direct age standardisation, reporting the ceiling effect where adding additional conditions had little impact on prevalence.** Labelled coloured lines represent multimorbidity prevalence calculated when considering different numbers of conditions in the count ranging from 2 to all 80 conditions stratified into IMD deciles where IMD 1 is least and IMD 10 is most deprived. Black dots represent the number of conditions at which RR >0.99 of multimorbidity prevalence of having the same multimorbidity prevalence when all 80 conditions were considered (ceiling effect): IMD 10 at 51 conditions, IMD 9 at 53 conditions, IMD 8 at 51 conditions, IMD 7 at 51 conditions, IMD 6 at 51 conditions, IMD 5 at 52 conditions, IMD 4 at 51 conditions, IMD 3 at 52 conditions, IMD 2 at 52 conditions, and IMD 1 at 54 conditions.
(TIFF)

**S2 Fig. Sex stratified multimorbidity prevalence according to number of conditions considered without direct age standardisation, reporting the ceiling effect where adding additional conditions had little impact on prevalence.** Labelled coloured lines represent multimorbidity prevalence calculated when considering different numbers of conditions in the count ranging from 2 to all 80 conditions stratified by sex. Black dots represent the number of conditions at which RR >0.99 of multimorbidity prevalence of having the same multimorbidity prevalence when all 80 conditions were considered (ceiling effect): women and girls at 54 conditions and men and boys at 51 conditions.
(TIFF)

## Acknowledgments

Dr. Niall Anderson for providing statistical consultancy.

## Author Contributions

**Conceptualization:** Clare MacRae, Bruce Guthrie.

**Data curation:** Megan McMinn, Bruce Guthrie.

**Formal analysis:** Clare MacRae, Megan McMinn, Stewart W. Mercer, David Henderson, Bruce Guthrie.

**Funding acquisition:** Clare MacRae, Stewart W. Mercer, David A. McAllister, Emily Jefferson, Daniel R. Morales, Bruce Guthrie.

**Investigation:** Clare MacRae.

**Methodology:** Clare MacRae, Ronan A. Lyons, Bruce Guthrie.

**Project administration:** Clare MacRae, Bruce Guthrie.

**Resources:** Clare MacRae.

**Software:** Clare MacRae.

**Supervision:** Stewart W. Mercer, Jane Lyons, Chris Dibben, Bruce Guthrie.

**Validation:** Clare MacRae, Bruce Guthrie.

**Visualization:** Clare MacRae.

**Writing – original draft:** Clare MacRae.

**Writing – review & editing:** Clare MacRae, Megan McMinn, Stewart W. Mercer, David Henderson, David A. McAllister, Iris Ho, Emily Jefferson, Daniel R. Morales, Jane Lyons, Ronan A. Lyons, Chris Dibben, Bruce Guthrie.

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
