## [Editor Report · Decision Letter 0]

14 Oct 2022

Dear Dr MacRae, 

Thank you for submitting your manuscript entitled "Measuring multimorbidity: impact of varying the number and selection of conditions on estimated multimorbidity prevalence in a large primary care population dataset" for consideration by PLOS Medicine.

Your manuscript has now been evaluated by the PLOS Medicine editorial staff as well as by an academic editor with relevant expertise and I am writing to let you know that we would like to send your submission out for external peer review.

Please re-submit your manuscript within two working days, i.e. by Oct 18 2022 11:59PM.

Kind regards,

Philippa Dodd, MBBS MRCP PhD

Senior Editor

PLOS Medicine

---

## [Decision Letter · Decision Letter 1]

14 Dec 2022

Dear Dr. MacRae,

Thank you very much for submitting your manuscript "Measuring multimorbidity: impact of varying the number and selection of conditions on estimated multimorbidity prevalence in a large primary care population dataset" (PMEDICINE-D-22-03375R1) for consideration at PLOS Medicine. 

[LINK]

In light of these reviews, I am afraid that we will not be able to accept the manuscript for publication in the journal in its current form, but we would like to consider a revised version that addresses the reviewers' and editors' comments. Obviously we cannot make any decision about publication until we have seen the revised manuscript and your response, and we plan to seek re-review by one or more of the reviewers. 

We expect to receive your revised manuscript by Jan 04 2023 11:59PM. Please email us (plosmedicine@plos.org) if you have any questions or concerns.

We look forward to receiving your revised manuscript. 

Sincerely,

Philippa Dodd, MBBS MRCP PhD

PLOS Medicine

plosmedicine.org

GENERAL

Please respond to all editor and reviewer requests detailed below, in full.

Please number the lines in the manuscript starting at 1 at “Abstract” and in continuous sequence thereafter.

We agree with the reviewers (please see below) that the study’s primary goal should be better emphasized/identified and its implications. Further details and clarifications are required throughout, which we agree with. Please revise accordingly.

** We agree that the novelty is a bit lost among the formal data reporting. Please revise in mind of the potential impact of the study outcomes at the patient/policy level **

COMMENTS FROM THE ACADEMIC EDITOR

I reviewed the article and the reviews and I agree with the decision for major revision

ABSTRACT

Please structure your abstract using the PLOS Medicine headings (Background, Methods and Findings, Conclusions).

Please combine the Methods and Findings sections into one section, “Methods and findings”.

Abstract Methods and Findings:

Please ensure that all numbers presented in the abstract are present and identical to numbers presented in the main manuscript text.

Please provide further details on the population (and setting) that the study investigates

Please provide further details as to how conditions were “selected” when considering multimorbidity prevalence (diagnostic coded registers?)

Please clearly define the main outcome measures.

Please quantify the main results with p-values as well as 95% CIs, if not for the purpose of transparent data reporting please provide the reasons why

Please include any important dependent variables that are adjusted for in the analyses.

Please include a summary of adverse events if these were assessed in the study.

In the last sentence of the Abstract Methods and Findings section, please describe the main limitation(s) of the study's methodology.

Abstract Conclusions:

Please replace the sub-heading “interpretation” with “conclusions”

Please address the study implications without overreaching what can be concluded from the data; the phrase "In this study, we observed ..." may be useful.

Please interpret the study based on the results presented in the abstract, emphasizing what is new and what the wider impact of the study outcomes are, without overstating your conclusions.

Please avoid vague statements such as "these results have major implications for policy/clinical care". Mention only specific implications substantiated by the results.

Please avoid assertions of primacy ("We report for the first time....")

AUTHOR SUMMARY

INTRODUCTION

Please see reviewer comments – we agree that limitations to the age of the data-set should be discussed, please do so later in the discussion 

METHODS and RESULTS

Page 4, data sources: please define SEP at first use in this section

Did your study have a prospective protocol or analysis plan? Please state this (either way) early in the Methods section.

For all observational studies, we ask authors to indicate the following clearly in the manuscript text: 

(1) the specific hypotheses you intended to test, 

(2) the analytical methods by which you planned to test them, 

(3) the analyses you actually performed, and 

(4) when reported analyses differ from those that were planned, transparent explanations for differences that affect the reliability of the study's results. If a reported analysis was performed based on an interesting but unanticipated pattern in the data, please be clear that the analysis was data-driven.

Please quantify the main results with p-values as well as 95% CIs, if not for the purpose of transparent data reporting please provide the reasons why

When a p value is given, please specify the statistical test used to determine it.

Please remove role of the funding source from the end of the methods section and include only in the manuscript submission form.

TABLES and FIGURES

We agree the reviewer that the figures rather undersell the data. The titles are rather inaccessible and difficult to understand and don’t explain the figure contents clearly for the reader. Please revise throughout, including the supporting files.

To make figures more accessible to those with colorblindness please consider avoiding the use of red/green 

Please ensure that each figure/table (including those in supporting files) has an appropriate caption

Please define all abbreviations (for example: IMD, RR) in the figure/table captions

Please indicate whether analyses are adjusted and if so, please state the caption inwhich factors are adjusted for and report unadjusted analyses for comparison

DISCUSSION

Please present and organize the Discussion as follows: a short, clear summary of the article's findings; what the study adds to existing research and where and why the results may differ from previous research; strengths and limitations of the study; implications and next steps for research, clinical practice, and/or public policy; one-paragraph conclusion. Please remove the sub-heading conclusion.

Please expand your discussion to better focus on the study aim and implication of the outcomes, without overstating – see also reviewer 2 comments

Please remove all “declaration” statements from the end of the discussion and include only in the submission form. The section “Author contributions” may remain under its own heading.

REFERENCES

Please use the "Vancouver" style for reference formatting, and see our website for other reference guidelines here: https://journals.plos.org/plosmedicine/s/submission-guidelines#loc-references

For in-text reference call outs citations should be placed in square brackets and preceding punctuation like so [1,2,3,4] or [1-4,6]. Please amend throughout.

In the bibliography please list up to but no more than 6 author names followed by et al (where more than 6 authors contribute to the study).

Journal name abbreviations should be those found in the National Center for Biotechnology Information (NCBI) databases. 

Comments from the reviewers:

Reviewer #1: Statistical review

This paper uses data from a large set of patients in England to investigate how different definitions of multimorbidity affect prevalence estimates. The authors also present this stratified by socioeconomic level, sex and age.

Generally the paper presents an important message for multimorbidity research. The statistical methods used are straightforward, as just prevalence estimates and confidence intervals are needed. I had very few comments, which are presented below: 

1. Statistical methods: "To examine the effect of the number of conditions considered in the count, multimorbidity prevalence was calculated when considering the most common two conditions, followed by the most common three conditions, the most common four conditions, etc, for every number up to considering all 80 conditions in the count (Information Panel). To do this, conditions were ordered from most to least prevalent (Supplementary Table 1) and added in turn to each new count" Although this procedure would give the highest multimorbidity prevalence in the case that conditions are independent from one another, it's possible it might not do so if common conditions are correlated. E.g. if considering conditions 1, 2 and 3 in descending prevalence, the prevalence of '1 and 2' might be less than '1 and 3' if conditions 1 and 2 co-occur in a lot of patients. This is just a picky point really and I don't think it makes sense to change what the authors did in any way!

2. Statistical methods: can some more intuition be provided about the direct standardisation to the age structure of the population?

3. Page 8: "29.5% (95%CI 29.5-29.9%)" - is this correct? The lower boundary and the point estimate are the same.

4. Discussion: are the prevalence estimates generally an underestimate due to potential under-diagnosis of some conditions?

James Wason

Reviewer #2: Thank you for the opportunity to review this work. In their manuscript, the authors explore how various definitions of multimorbidity influence its estimated prevalence. As the authors acknowledge, the growing attention to multimorbidity - and its highly varied operationalization - has contributed to methodological variation and biases in the literature. Overall, this manuscript covers an important aspect of multimorbidity research.

I have a few comments for the authors to consider. Overall, though, I think the paper would benefit from identifying the primary goal and really tightening up the methods, results, and discussion around this. It was, at times, a bit hard to follow. I am not advocating for removing any of the current analyses, but rather making sure they are presented in a way that easily conveys the overarching theme and story while highlighting the important and novel aspects of this work.

Major:

1. Introduction: the authors state "it is common for authors not to explain the procedure used to decide which conditions they consider in the count" - I don't think this is necessarily true. Instead, there's often a 'recycling' of previously published lists. This could be reframed to highlight that the heterogeneity the authors are exploring is not due to authors not presenting information.

2. The authors acknowledge in the introduction that the relationship between multimorbidity and time is important to note given its growing prevalence. However, the data the authors analyze is now nearly 7 years old. While I doubt it would make any meaningful change in the results, the authors should consider acknowledging the datedness of these data and whether they believe this influences their findings at all.

3. Would the authors consider making Supplemental Table 1 part of the main text? I think the list of included conditions is the crux of this study. I know it's visually unappealing, but I think could be important.

4. Could the authors, in the methods, more clearly state that they included every condition mention in any of those studies? It seems to me that is what the authors did, but I think this needs to be made explicitly clear.

5. The approach to use a relative risk and evaluate the appropriate 'cutoff' given those 80 conditions is an interesting approach but could be better setup in the introduction and the methods. It's a bit confusing, right now, the various comparisons that are being made. Indeed, I'd encourage the authors to consider an approach where they explicitly and succinctly lay out all their analyses. For example, "We conducted a suite of comparisons including: ____."

6. The relative risk analysis seems to be the most novel part of this studied but is not as prominently featured. I would encourage the authors to restructure their manuscript focusing on this if they also believe it's the most novel. Right now, some of the most interesting findings feel a bit buried in the analysis.

a. Similarly, it's unclear what the intermediary analyses contribute. For example, "When considering only the two conditions…" as presented in the results. While I understand this was a step along the way, I'm not sure that any study would ever pick only two conditions to estimate the prevalence of multimorbidity.

b. Additionally, I think the first sentence of the discussion doesn't highlight the most important findings. Of course, when it's 2 versus 80 there is a large difference. There are several other findings that are better to highlight - for example the sentence beginning "Second, multimorbidity…"

Minor:

1. The authors could remove the date from the abstract methods.

2. The abstract findings are hard to follow without the additional context presented in the full manuscript. I would encourage the authors to try to restructure their findings to make it more interpretable.

3. Could the authors instead spell out "November 30, 2015" since DD/MM/YY is not standard globally.

4. Please mention the funders in the methods.

5. Please define "IMD" in Table 1.

Reviewer #3: Dear editor, 

Thank you for allowing me to read such an interesting manuscript. 

I believe that the manuscript was well written and that the research was presented in a way that was easy to follow and understand. I think the statistics were adequately explained and their meaning was easy to understand. 

I have only one question for the authors: what were the criteria for choosing some condition lists like the ones from Diederichs, Fortin, Ho, Barnett, Salive, but not others that are well know such as O'Halloran and N'Goran (particularly in primary care settings)?

- O'Halloran J, Miller GC, Britt H. Defining chronic conditions for primary care with ICPC-2. Fam Pract. 2004 Aug;21(4):381-6. doi: 10.1093/fampra/cmh407. PMID: 15249526.

- N'Goran AA, Blaser J, Deruaz-Luyet A, Senn N, Frey P, Haller DM, Tandjung R, Zeller A, Burnand B, Herzig L. From chronic conditions to relevance in multimorbidity: a four-step study in family medicine. Fam Pract. 2016 Aug;33(4):439-44. doi: 10.1093/fampra/cmw030. Epub 2016 May 6. PMID: 27154549.

Best regards.

[LINK]

---

## [Decision Letter · Decision Letter 2]

8 Feb 2023

Dear Dr. MacRae,

Thank you very much for re-submitting your manuscript "Measuring multimorbidity: impact of varying the number and selection of conditions on estimated multimorbidity prevalence in a large primary care population dataset" (PMEDICINE-D-22-03375R2) for review by PLOS Medicine.

I have discussed the paper with my colleagues and the academic editor and it was also seen again by 3 reviewers. I am pleased to say that provided the remaining editorial and production issues are dealt with we are planning to accept the paper for publication in the journal.

[LINK]

We look forward to receiving the revised manuscript by Feb 15 2023 11:59PM.   

Sincerely,

Philippa Dodd, MBBS MRCP PhD

PLOS Medicine

plosmedicine.org

Requests from Editors:

GENERAL

Thank you for your detailed and considerate responses to previous editor and reviewer requests. Please see below for further minor revisions.

DATA AVAILABILITY STATEMENT

Thank you for making your data available. 

A point for clarification - in one section of the manuscript submission form you state that some restrictions will apply to your data availability but subsequently you go on to state that all relevant data are included in the manuscript and supporting files. I have detailed our data availability statement below for your information, please clarify/revise/expand your statement as necessary.

PLOS Medicine requires that the de-identified data underlying the specific results in a published article be made available, without restrictions on access, in a public repository or as Supporting Information at the time of article publication, provided it is legal and ethical to do so. Please see the policy at 

http://journals.plos.org/plosmedicine/s/data-availability

and FAQs at 

http://journals.plos.org/plosmedicine/s/data-availability#loc-faqs-for-data-policy

PLOS defines the “minimal data set” to consist of the data set used to reach the conclusions drawn in the manuscript with related metadata and methods, and any additional data required to replicate the reported study findings in their entirety. Authors do not need to submit their entire data set, or the raw data collected during an investigation. Please submit the following data:

The values behind the means, standard deviations and other measures reported;

The values used to build graphs;

The points extracted from images for analysis.

For each data source used in your study: 

TITLE

Suggest the following to align better with PLOS Medicine’s preferred style, 

The impact of varying the number and selection of conditions on estimated multimorbidity prevalence: a cross sectional study using a large, primary-care population dataset

STATISITICAL REPORTING

Thank you for including p values. Please ensure that throughout the manuscript including in the tables, figures and supplementary files p is reported as p <0.001 (not .001 or .0001) and where higher as p=0.002 (not .002), for example. 

Throughout, suggest reporting statistical information as follows: X% (95%CI [X,Y] p <0.001)

 - Note the use of square parentheses around upper and lower confidence limits

 - Note the use of a comma instead of a hyphen (as these can be confused with negative values) to separate upper 

 and lower limits

 - Note the reporting of p values as detailed above

ABSTRACT

Line 30: suggest “We conducted….” Or something similar

Line 32: sentence beginning “Outcome…” is rather long and could be more accessible, please revise

Line 35: please ensure that HDR[-UK] is defined for the reader 

Lines 38 - 46: the word is “prevalence is mentioned a number of times, might it be helpful to remind the reader that you are reporting multimorbidity prevalence at some point, we leave it to your discretion bt it may be worth considering.

Line 42: is the word “prevalence” required twice here or is this a typo? Please clarify/revise 

Line 54 - 55: we suggest avoiding naming the lists here as the reader is yet to learn what all the lists are. See comments below also. Suggest, “These findings imply that there is a need for a standardised approach to defining multimorbidity.” Then go on to state (concisely) how your study helps to solve this problem. 

AUTHOR SUMMARY

Line 64: “in half of studies…” suggest make into a separate bullet point

Line 67: suggest “2” instead of “two to 285 (median 17).”

Line 68: perhaps one line stating why? i.e. comparability of study outcomes, generalisability etc

Line 72: suggest “and the impact on multimorbidity prevalence” perhaps?

Line 73: It would be helpful for the reader to elaborate here – what were the differences, briefly

Line 75: this won’t mean anything to the reader as it hasn’t been introduced anywhere in the text apart the abstract but without adequate context (see above for a similar point) suggest either removing/modifying this statement or revising the summary (and abstract) accordingly such that it’s clear to the reader what this is/means. For example, you could expand bullet point 1 of this sub-section to include (some examples) of the lists you used to derive counts and then go on to state which of these were most reliable, as we understand things. 

Line 83: suggest removing this statement as it is best placed as part of a discussion

** The early part of your introduction, description of study design in your methods section and, the early part of your discussion summarise things very well and may be helpful with the revisions to your author summary **

REFERENCES

Please ensure that punctuation follows, not precedes, in-text reference call-outs. For example, line 94, “…systems worldwide.[1]” should read, “…systems worldwide [1]. Please check and amend throughout

SOCIAL MEDIA

To help us extend the reach of your research, please provide any Twitter handle(s) that would be appropriate to tag, including your own, your coauthors’, your institution, funder, or lab. Please detail any handles you wish to be included when we tweet this paper, in the manuscript submission form when you re-submit the manuscript.

Comments from Reviewers:

Reviewer #1: Thank you to the authors for addressing my previous comments well. I have no further issues to raise.

Reviewer #2: The authors have thoughtfully addressed the points raised. Thank you.

Reviewer #3: The authors have responded to the remaining comments made by me.

Thank you and best wishes.

[LINK]

---

## [Editor Report · Decision Letter 3]

20 Feb 2023

Dear Dr. MacRae,

Thank you very much for re-submitting your manuscript "The impact of varying the number and selection of conditions on estimated multimorbidity prevalence: a cross-sectional study using a large, primary care population dataset" (PMEDICINE-D-22-03375R3) for review by PLOS Medicine.

I have discussed the paper with my colleagues and I am pleased to say that provided the remaining editorial and production issues are dealt with we are planning to accept the paper for publication in the journal.

[LINK]

We look forward to receiving the revised manuscript by Feb 23 2023 11:59PM.   

Sincerely,

Philippa Dodd, MBBS MRCP PhD

PLOS Medicine

plosmedicine.org

Requests from Editors:

GENERAL

Thank you for your responses to previous editor requests.

Your author summary needs further revision before we can publish your manuscript. Please see below

AUTHOR SUMMARY

* What did the researchers do and find – this section should be no more than 4 bullet points.

Without reading other parts of the manuscript, it’s difficult to understand what you did. Further some points, to me seem to be illogically ordered. Please see specific points below and revise accordingly.

Line 74 and line 78: suggest combining (and shortening) these points into one point with additional details about what the “lists” are, for example, “We examined nine published condition lists used to define and measure co-morbidity, multi-morbidity and its prevalence. We combined different numbers and types of conditions (detailed in these lists?) to determine how multimorbidity prevalence changed with varying classifications/definitions… All conditions were counted in the same way….” or something similar 

The statement currently at line 76 should then follow the above

Line 84: should precede that at line 81

** What do these findings mean – this section should explain the implications of your study findings only.

Line 90: The first point here justifies your methodology and seems inappropriately placed. The sentence is also rather long. Please remove this. If you think this point is important to make, then it should be made in the “what did the researchers do and find” section (see above).

Line 96: When reading the full summary, it suggests that using lists which report “highest” multimorbidity prevalence is advisable. Please (concisely) elaborate as to why this is the case – do these yield more consistent results (?), for example

[LINK]

---

## [Editor Report · Decision Letter 4]

27 Feb 2023

Dear Dr MacRae, 

On behalf of my colleagues and the Academic Editor, Professor Aaron Kesselheim, I am pleased to inform you that we have agreed to publish your manuscript "The impact of varying the number and selection of conditions on estimated multimorbidity prevalence: a cross-sectional study using a large, primary care population dataset" (PMEDICINE-D-22-03375R4) in PLOS Medicine.

PRESS

Thank you again for submitting to PLOS Medicine, it has been a pleasure handling your manuscript. We look forward to publishing your paper. 

Best wishes,

Pippa 

Philippa Dodd, MBBS MRCP PhD 

PLOS Medicine